# Anthocyanins: Potential Therapeutic Approaches towards Obesity and Diabetes Mellitus Type 2

**DOI:** 10.3390/molecules28031237

**Published:** 2023-01-27

**Authors:** Denise Franco-San Sebastián, Samary Alaniz-Monreal, Griselda Rabadán-Chávez, Natalia Vázquez-Manjarrez, Marcela Hernández-Ortega, Gabriela Gutiérrez-Salmeán

**Affiliations:** 1Centro de Investigación en Ciencias de la Salud (CICSA), Facultad de Ciencias de la Salud, Universidad Anáhuac México, Huixquilucan 52786, Mexico; 2Instituto para la Investigación en Obesidad, Instituto Tecnológico y de Estudios Superiores de Monterrey, Campus Guadalajara, Guadalajara 64849, Mexico; 3Unidad de Metabolómica, Dirección de Nutrición, Instituto de Ciencias Médicas y Nutrición Salvador Zubirán, Ciudad de Mexico 14080, Mexico

**Keywords:** anthocyanins, diabetes, functional foods, inflammation, phytochemicals

## Abstract

Overweight and obesity are present in about three-quarters of the adult population in Mexico. The inflammatory mechanisms subjacent to visceral white adipose tissue are accountable for the initiation and development of cardiometabolic alterations, including type 2 diabetes. Lifestyle changes are pillars within its therapeutics and, thus, current dietary modifications should include not only hypocaloric prescriptions with balanced macronutrient intake, preferably by increasing the amount of whole grains, fruits, vegetables, nuts and legumes, but in concomitance, bioactive substances, such as anthocyanins, have been correlated with lower incidence of this disease.

## 1. Introduction

The prevalence of overweight and obesity in the Mexican population is about 75.2% in adults aged 20 years and older. As defined by the World Health Organization, obesity is a chronic disease characterized by excess adiposity which has negative impacts on health, as fat mass is metabolically active. This increases adipose tissue, particularly denominated visceral fat that results in low-grade sustained inflammation which drives the development of cardiometabolic and chronic degenerative disorders, such as diabetes, with an implication for the quality of life of individuals, families, and society worldwide. According to the statistics of the International Diabetes Federation from 2017, the global prevalence of diabetes in the age group of 20 to 79 years is 8.8% [1], and is considered the fifth leading cause of death worldwide. The number of patients with diabetes around the world in 2014 was 422 million, and it is estimated that by 2035, it will be 592 million [2]. In Mexico, it is considered a public health problem. A total of 14.4% of adults over 40 years of age have this disease, and this percentage increases to 30% in those over 50 years of age. According to data from the National Survey of Health and Nutrition (ENSANUT) in 2016, the number of the people aged 20 and over with a previous medical diagnosis of diabetes increased from 6.4 million adults living with diabetes in 2012 to 8.6 million in 2018. These data make up an alarming situation in which more efficient measures should be focused on the prevention and treatment of diabetes. Current dietary intervention strategies are based on lower caloric intake and a balanced diet, especially through the intake of whole grains, fruits, vegetables, nuts, and legumes There is considerable evidence that suggests the consumption of whole grains in the diet as a healthy alternative [3]. They provide fiber, vitamins, minerals, and phytochemicals, such as polyphenols, carotenoids, and phytosterols [4]. They reduce the risk of cardiovascular disease, diabetes mellitus, obesity, and certain types of cancer [5]. The benefits associated with a healthy diet may be related to the presence of unsaturated fats, dietary fiber, and certain phytochemicals that are contained in particular foods. However, other substances interact within the organisms in a more pathway-specific way (e.g., iminosugars (miglitol, voglital) compete with saccharides for the digestion site, thus decreasing its absorption. Other bioactive substances have been reported to modulate other metabolic pathways regarding glucose homeostasis [6] For example, anthocyanins (ANC) have been correlated with lower incidence of type 2 diabetes in humans [7]. Herein we present a narrative review on this topic.

## 2. Phytochemicals

These are, etymologically, compounds produced by plants. They are synthesized as secondary metabolites (i.e., they are not required for vital functions, such as growth, but confer an advantage to the organism). For example, capsaicin helps chili peppers defend themselves from frugivorous predators. Among these phytochemicals, polyphenols have been of recent interest. 

Polyphenols are a vast group of organic compounds that have one or two aromatic rings with multiple hydroxyl groups. The most studied and known function of these phenols is their antioxidant activity, as they deactivate oxygen, and are free radical hydrogen donors, providing a protective effect on cells against oxidative damage [8]. However, polyphenols are categorized into five groups: flavonoids, stilbenes, phenolic acids, coumarins and tannins. Within the first group, we find anthocyanins: water-soluble flavonoids which are responsible for the red, violet, and blue color in fruits, vegetables, and cereal grains [9].

### Structure, Classification, and Function of Anthocyanins

Anthocyanins belong to the group of flavonoids, and their basic structure is a flavone nucleus, which consists of two aromatic rings linked by a three-carbon unit. Anthocyanins are synthesized through the phenylpropanoid pathway, which initiates from phenylalanine (and tyrosine in monocotyledonous plants). This amino acid is metabolized to cinnamic acid, which is further used for chalcone production: chalcones are the precursors to all flavonoids. The level of hydroxylation and methylation in the “B” ring of the molecule determines the type of anthocyanidin. Anthocyanidin (aglycone form) is the central structure of anthocyanins [10]. The addition of a sugar chain results in the glycosidic form of the anthocyanidin molecule, called anthocyanin [9]. There are 23 anthocyanidins and 500 anthocyanins isolated in plants. Their differences are mainly due to the degree of methylation, which influences the stability of the molecule and the hue; that is, the greater the methylation, the greater the red pigmentation and stability [10]. Anthocyanin biosynthesis is shown in Figure 1. The degree of hydroxylation and the number and nature of the chains attached to the sugar residues are also important in their classification, the most common of which being glucose, ribose, xylose, and galactose [11]. As shown in Figure 2, there are six main compounds of anthocyanidins in food: 50% of anthocyanins from fruits and cereals (12%) are derived from cyanidin, followed by pelargonidin, delphinidin (12%), peonidin (12%), petunidin (7%), and malvidin (7%) [10]. The stability of anthocyanin molecules depends on factors, such as temperature, pH, exposure to light, oxygen, sulfites, and ascorbic acid [12,13].

Plants use anthocyanins against extreme temperatures, so they are found in the outer layers of the cell, the epidermis and the mesophyll, and most flowers and leaves, as well as within the fruits.

Some foods high in anthocyanins are strawberries, blueberries, black currant, currants, and raspberries, with a range of 100–700 mg/100 g. Other good sources of anthocyanins can include blue corn, plums, pomegranate, eggplant, wine, grapes, and red or purple vegetables, such as red cabbage, black carrot, and purple cauliflower [14].

#### Anthocyanins in Pigmented Corn as an Example of Functional Food

Mexico is considered to be the place of the origin, domestication, and diversification of corn, or maize. Zea mays, its scientific name, is an annual grass native to Mesoamerica. Its domestication began approximately twelve thousand years ago in the neovolcanic axis of Mexico, and it was introduced to Europe in the 16th century. There are between 59 and 64 types of corn in the Mexican territory. Cornstarch isolate is the most widely used ingredient [15]. The type of corn and its subspecies depend on the amount and type of starch in the grain [16]. There is a great genetic variety in the color of corn grains [17], including black, blue, pink, red, purple, and brown. In purple corn, the anthocyanins are found in the peri-layer; in blue corn, the anthocyanins are found in the aleurone layer. In blue, pink, and scarlet maize, 15 anthocyanins have been identified, and in red maize there are 27 different anthocyanins. In general, the most identified anthocyanin is the 3-glucoside cyanidin [18], but many aglycones have also been identified, including cyanidin, peonidin, and pelargonidin. In general, the varieties of corn with color contain between 27 to 1439 mg of anthocyanins/kg in dry grain [19], due to the fact that the mature grains are the ones with the highest anthocyanin content (indicating that anthocyanins are formed in that stage), and the degree of maturity during harvest impacts the anthocyanin content [20,21].

## 3. Mechanisms of Absorption, Metabolism, and Bioavailability of Polyphenols

The absorption and metabolism of polyphenols has been studied extensively. The biochemical pathways, related to their bioavailability, are well understood in most classes of polyphenols. In humans, however, this can be somewhat complicated due to the various metabolic pathways and their interactions with the gut microbiota. Its absorption is also influenced by factors, such as pH, enzymatic activity, or body temperature [22]. The digestive process begins in the mouth when the anthocyanins come into contact with saliva and are partially degraded (about 60%) under conditions of temperature (generally 37 °C) and pH (6.4) in approximately one hour. In ex vivo studies, glycosylated delphinidin and petunidin have been shown to degrade more rapidly than cyanidin, pelargonidin, malvidin, and peonidin under oral conditions. Conjugated anthocyanins are more stable than monosaccharides, and less degradation has been observed when there are fewer bacteria in the oral microbiota. Anthocyanins pass quickly through the oral cavity, so their real degradation corresponds to less than 5% in this stage of digestion, allowing anthocyanins to travel to the stomach [23]. In the gastric phase, the food is broken down into smaller parts so that the nutrients are absorbed and metabolized. It has been suggested that anthocyanins are rapidly absorbed in the stomach and intestine, reaching their maximum concentration in the blood 1 h after ingestion. Its greater absorption in the stomach is due to the stability it acquires in acidic pH (pH: 2) [24]. 

It is generally estimated that the absorption area of anthocyanins in the stomach is approximately more than 20% [25]. In the gastric mucosa, anthocyanins bind to bile translocase (an organic anion transporter), and this protein favors its diffusion and absorption in the circulatory system after passing through the liver [26]. Some structures have been explained in vivo in detail, such as quercitin, hesperidin, and some phenolic acids. The degree of absorption is also influenced by several factors, such as the chemical coupling of the polyphenol [27], the solubility, and the process, as well as the fat content. 

Peak concentrations of postprandial blood polyphenol are generally less than 1 μM, while for intestinal catabolites, concentrations can exceed this figure and are usually 10 to 100 times higher than the original compounds. Most of the circulating metabolites both in the original compound and in the intestine are in the form of glucuronides or sulfates, and may also be methylated [28]. Classical bioavailability studies in mice show that the consumption of epicatechin, its absorption and urinary excretion is 80% as shown in an investigation from Otaviani JL in 2016. The concentration reached by some polyphenols in the blood depends largely on the parent polyphenol administered, and the concentrations of metabolites in the gut microbiota are likely to be higher than in plasma [29,30]. 

In the liver, polyphenols are metabolized by methylation and glucuronidation reactions. The glycosylated anthocyanins that are absorbed continue to the small intestine, the absorption capacity of which varies from 10.7%, for example, in malvidin 3-glucoside, to 22.4% in cyanidin 3-glucoside. The small intestine is considered as a fast and efficient absorption site, although it has been seen that the anthocyanins in this site reduce its bio accessibility by 75% [31]. Two mechanisms have been proposed for the absorption of anthocyanins in the intestine. One is by passive diffusion through the membrane of epithelial cells, since the aglycones of anthocyanins have great hydrophobicity. It has also been suggested that they may leak into the bloodstream as bioavailable anthocyanin products, such as hydroxybenzoic acids, derived from the B ring. Another process is from the sodium–glucose cotransporter at the brush border of enterocytes. The enterocyte absorbs 50% of the anthocyanins. Despite these proposals of absorption mechanisms, it is estimated that only 1% of the anthocyanins will reach the blood circulation [26].

The detection of anthocyanin metabolites may be due to different reactions that conjugate flavonoids in the intestine and liver. The metabolites present in plasma may also be due to the enterohepatic cycle, where they are absorbed and transported by hepatocytes and subsequently excreted with bile. This can cause a second absorption of these compounds to take place in the being [25].

In the large intestine, a large amount of anthocyanins pass, intact, to the colon and are metabolized by colonic bacteria as phenolic compounds, which contributes to their antioxidant activity. The microbiota provides a wide variety of enzymes that hydrolyze phenolic compounds, forming bacterial metabolites.

## 4. Chemical Nature, Reported Mechanism of Action and Assigned Effects

The high consumption of polyphenols, such as anthocyanins, has been related to a lower incidence of type 2 diabetes in humans [27]. The reduction in glucose absorption in the intestine is desirable in patients with hyperglycemia and type 2 diabetes, since high concentrations in the bloodstream are associated with greater comorbidities. 

Recent studies (in vitro and in vivo) have revealed the beneficial glucose control properties of anthocyanins [32]. Some extracts rich in anthocyanins from different sources, such as black currant, red raspberries, purple rice, and black soybeans, have been shown to have hypoglycemic abilities [31,32,33,34,35,36]. In particular, blackberry extract can suppress hepatic gluconeogenesis and modulate glucose metabolism in human HepG2 cells [37]. However, there are few studies available related to the increase in insulin secretion induced by anthocyanins through food sources [38].

Foods such as pigmented corn contain anthocyanins, especially purple corn, which has shown promising antidiabetic activities [32]. Recent studies have shown that the anthocyanin-rich extract from the peri-layer of purple corn decreases insulin resistance and increases glucose uptake in 3T3-L1 adipocytes [39]. Among the mechanisms of action of greatest scientific interest in glycemic control, the receptors stand out: FFAR1, also known as GPR40 (free fatty acid receptor), and GK (glucokinase receptor) [40].

FFAR1 activation is associated with decreased fasting blood glucose and improved glucose tolerance in diabetic rats [41]. The GK receptor is expressed on pancreatic β-cells and hepatocytes. It plays a crucial role in glucose homeostasis. Studies in rodents with insulin resistance or type 2 diabetes have lower hepatic expression of the GK receptor, suggesting an alteration in the regulation of this biomarker [42]. Various studies have reported the impact of polyphenols on these two receptors [43]. However, evidence has been lacking for the interaction of anthocyanidins from dietary sources and their relationship with receptors (FFAR1/GK) with insulin secretion and glucose absorption. The study conducted by Diego A. Luna Vital et al. demonstrated that blue corn anthocyanins reduce epithelial glucose transport, determining that the highest concentrations of pure anthocyanins (100 μM) were sufficient to reduce glucose transport in epithelial cells with a reduction range from 15 to 27%. In this same study, it was observed that purple corn anthocyanin-enriched water extract (PCW) abruptly reduced glucose transport concentration by 33% at a concentration of 0.5 mg/mL. Another of the main mechanisms described in the reduction in glucose transport through the effect of anthocyanins is the interaction with the glucose transporter (GLUT2) and the sodium-glucose transporter 1 (SGLT1) [28].

Purple corn anthocyanins increase insulin secretion glucose-stimulated (GSIS) by modulating FFAR1 in pancreatic β-cells. A study by Suantawee et al. reports that cyanidin stimulates insulin secretion in iNS-1E cells by activating type 1 voltage-gated calcium channels [44]. However, anthocyanidins must have different types of action since anthocyanidins and anthocyanins have different bioactivities [45].

Purple corn anthocyanidins modulate protein expression of the FFAR1 insulin secretion-dependent pathway. PCW is the most efficient treatment, increasing FFAR1 expression by 100% compared to the untreated control group [39]. Purple corn anthocyanins increase glucose reuptake and activate the GK receptor on liver cells. PCW is the most effective treatment by increasing glucose reuptake by approximately 45% of the pure anthocyanidins, and D3G (delphinidin) and C3G-P (cyanidin) were the most effective in increasing glucose reuptake, with 35% and 41%, respectively. The studies by Luna-Vital et al. open the possibility that the use of anthocyanins from purple corn activate FFAR1 and GK, markers that, when activated, decrease the complications of type 2 diabetes. For the first time it has been reported that anthocyanins from purple corn increase insulin secretion in cells β-pancreatic cells in vitro, potentially via FFAR1 activation [7].

In addition, purple corn anthocyanins increase hepatic glucose reuptake in vitro, which may contribute to glucose homeostasis. In general, anthocyanin-enriched extracts are more effective than pure anthocyanins. Purple corn anthocyanins are good candidates to be incorporated into the diet of patients with type 2 diabetes in their treatment.

### 4.1. Reduction of Adipogenesis

Several purple maize genotypes regulate proinflammatory agents in vitro. Particularly, these compounds reduce adipogenesis by regulating proinflammatory pathways and inhibiting fatty acid activity. Phenolic compounds also play a role in decreasing insulin resistance by inhibiting IRS-1 (inhibition of phosphorylation) and promoting the translocation of glucose transporters in the adipocyte membrane. In vivo studies have shown that extracts enriched with phenolic compounds prevent obesity in animal models by suppressing adipogenesis, gluconeogenesis, and lipogenesis [46].

### 4.2. Antioxidant Properties

Several studies have shown the antioxidant capacity of purple corn, which correlates in particular with the amount of the bioactive compound. Cevallos et al. concludes that the antioxidant activities of purple corn through anthocyanins are superior compared to other phenolic compounds by showing that purple corn phenolic compounds have a higher antioxidant capacity compared to blueberry phenolic compounds, suggesting that purple corn may have a greater activity of the hydroxyl groups and a more favorable configuration to interact with free radicals [47]. The antioxidant capacity of purple corn remains after the industrial process. The antioxidant properties of tortillas, chips, and grains of corn have been investigated using ORAC methods. The loss of anthocyanins after nixtamalization reduces the antioxidant capacity by approximately 28%, 37% in processed tortillas, and 55% in chips [48].

### 4.3. Anti-Inflammatory Activity

Purple corn pigments can weaken mesangial inflammation (induced by high glucose concentrations), expansion, and hyperplasia by inhibiting the inflammatory action of IL-8. Cells that receive the anthocyanin-enriched extract from purple corn decrease IL-8 secretion, depending on the dose level. Cyanidin is the most abundant anthocyanin in purple corn, and numerous studies have shown its anti-inflammatory properties. Cyanidin suppresses the inflammatory response, so tumor necrosis markers were significantly suppressed by cyanidin administration [49].

## 5. Discussion

As shown in the Table 1, several clinical studies have shown that anthocyanin supplementation significantly reduces TC, TG, and LDL levels in patients with dyslipidemia, and increases HDL cholesterol. In other systematic reviews and meta-analyses, they conclude with an improvement in glucose parameters, IL-6, as well as in the response of the absorbance capacity of oxygen radicals, determined by the areas under the curve and postprandial concentrations. However, the doses of anthocyanins include a very wide range (from 32 mg to 600 mg), and the form of consumption and the time of consumption is very variable due to the vehicular presentation and food sources. Moreover, this phytochemical consumption is not achieved within a normal daily basis, i.e., from diet, thus it is mandatory to provide bioactive substances in the form of nutraceuticals. This may have implications for the bioactive properties of anthocyanins through chemical and physical interactions, such as increased antioxidant capacity when anthocyanins are consumed in water or in combination with high-sugar foods or fasting, which is related to delayed absorption. There is no significant evidence that using higher doses of anthocyanins compared to lower doses has a more effective result; this phenomenon, known as homeorhesis, has been described elsewhere, but in brief, phytochemicals exhibit a bell-shaped dose–response effect. Thus, increasing the amount of their consumption does not necessarily reflect greater clinical benefits, and, rather, these may be lost. Most of these studies are based on the use of red fruits, in which it is important to keep in mind that a portion of 100 g of blueberries contains approximately 170 mg of anthocyanins, which can be easily implemented in a dietary strategy; nevertheless, other food sources—such as ancestral maize, taro leaves, purple banana, etc., should be studied, particularly in native environments, since their cost is likely to be significantly lesser. It seems that the most robust scientific evidence is found in animal studies, and that the evidence in humans is beginning to be attractive, but that it should definitely be deepened with long-term studies. The proposed mechanism of anthocyanins on diabetes mellitus type 2 and obesity are resumed in Figure 3 and Figure 4.

## 6. Conclusions

The consumption of fresh foods containing bioactive compounds should be promoted, since they provide health benefits. Particularly, plant-based foods contain several bioactive compounds, such as polyphenols, which modulate different pathways with antioxidant, anti-inflammatory activity, which, in turn, modulates glucose regulation, as well as neuroprotective activity. Among polyphenols, anthocyanins are an interesting group, from the sensory characteristics of food (color) to their great potential as nutraceutical ingredients for their health benefits, as they have been proven to attenuate body weight, glucose, and triglycerides via AMPK and TLR pathways, thus reducing adipogenesis and inflammation in adipocytes.

Adequate daily intake of anthocyanins promotes protection against chronic degenerative diseases. This can be promoted simply by frequent consumption of red fruits, pigmented vegetables, pigmented corn, or products made from pigmented corn in our daily diet. New formulations, such as encapsulated or nanoparticle anthocyanins, could provide greater stability of anthocyanins for use in nutraceuticals. These formulations could improve the bioavailability, improving the health benefits. However, the simplest way is by consuming a higher plant-derived diet that can be easily implemented as a public health strategy.

## Figures and Tables

**Figure 1 molecules-28-01237-f001:**
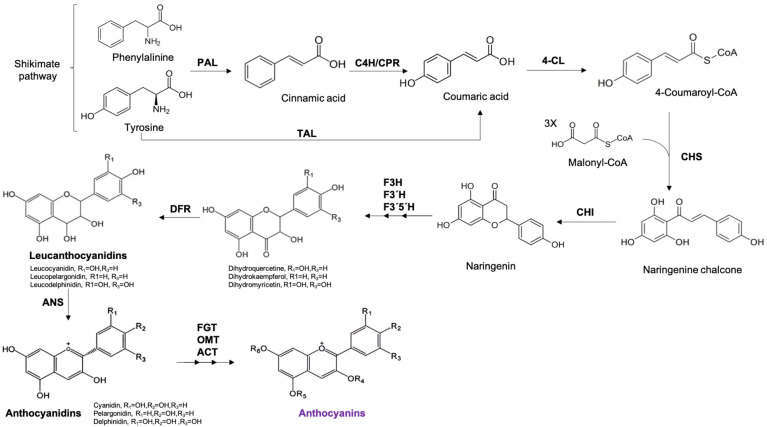
Biosynthesis pathway of anthocyanins in plants. PAL: Phenylalanine Ammonia Lyase, C4H: Cinnamate 4-hydroxylase, CPR: cyto-chrome P450 reductase, 4-CL: 4-coumaroyl-CoA ligase, CHS: Chalcone Synthase, CHI: Chalcone Isomerase, F3H: Flavanone-3-Hydroxylase, F3′H: flavonoid 3′-hydroxylase, F3′5′H:Flavonoid 3′,5′-hydroxylase, DFR: Dihydroflavonol 4-reductase, ANS: Anthocya-nidin Synthase, FGT: Flavonoid Glucosyltransferase, OMT: O-Methyltransferase, ACT: Anthocyanin Acyltransferase.

**Figure 2 molecules-28-01237-f002:**
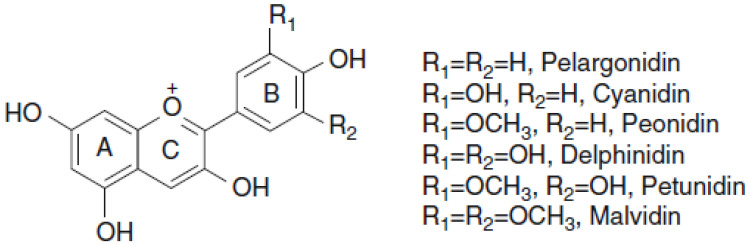
Anthocyanidin structures.

**Figure 3 molecules-28-01237-f003:**
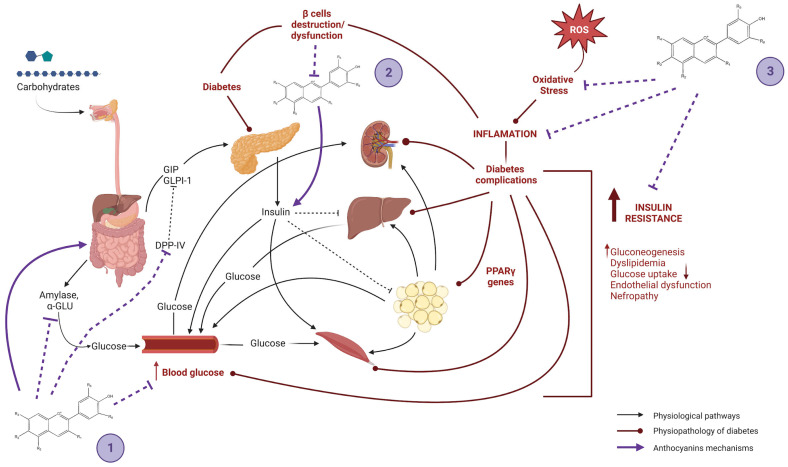
Anthocyanins effects on diabetes by regulation of incretin levels and dysbiosis, reduction in hyperglycemia and inhibition of digestive enzymes (1); cytoprotecting of pancreatic cells and regulation of insulin levels (2); and decreasing diabetes complications by reducing insulin resistance, inflammation, and oxidative stress (3). Created with BioRender.com.

**Figure 4 molecules-28-01237-f004:**
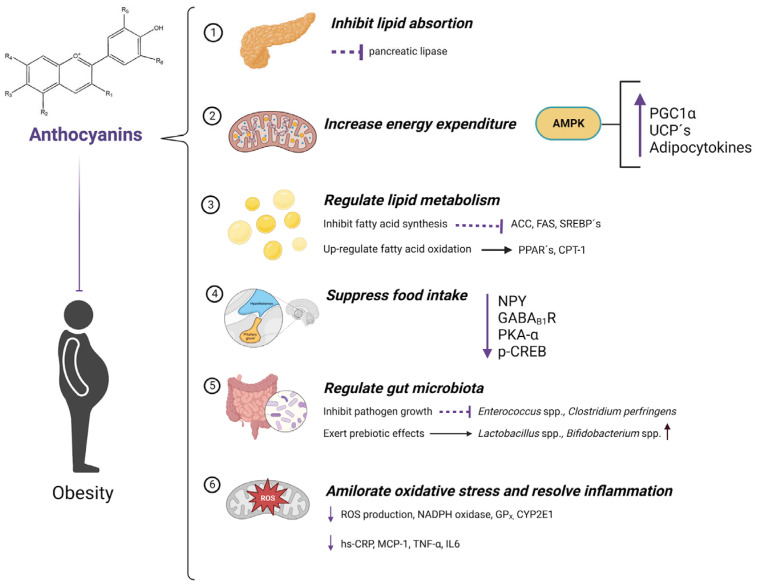
Molecular mechanisms of anti-obesity effects of anthocyanins. (1) Various anthocyanins are strong pancreatic lipase inhibitors; (2) These pigments can up-regulate AMPK expression which increases the expression of PGC1α and UCP’s, and the excretion of adipocytokines; (3) Anthocyanins inhibit fatty acid synthesis by down-regulating the expression of ACC, FAS, and SREBP’s and by up-regulation of fatty acid oxidation by increasing the expression of PPAR´s and CPT-1; (4) Pigments, such as anthocyanins, suppress food intake by reducing the expression of NPY, GABAB1R, PKA-α, and p-CREB; (5) Anthocyanins can inhibit growth of Enterococcus spp. and Clostridium perfringens, pathogens, and exert prebiotic effects enhancing the growth of beneficial strains, such as *Lactobacillus* spp. and/or *Bifidobacterium* spp.; (6) These pigments can ameliorate oxidative stress by reducing ROS production, inhibit expression of NADPH oxidase and GPx, as well as reducing the expression of CYP2E1, and also can resolve inflammation by decreasing levels of hs-CRP, MCP-1, TNF-α, and IL6.

**Table 1 molecules-28-01237-t001:** Analysis of studies and dosage and discussion of evaluated clinical studies.

Author	Study Design	Population	Dosage	Study Duration	Main Findings
María S. Hershey 2020 [50]	Cross-sectional ECA	249 men Age: 47 ± 7.6	Range from 6.05 to 120.8 mg/per ration	2 years	High intake of anthocyanins is associated with > HDL.
Zhu, 2013 [51]	Randomized, double-blind, parallel	73 participants, per group Age: 40–65	320 mg/day	24 weeks	Anthocyanin supplementation decreased TC, TG, LDL compared to placebo > HDL levels.
Soltani, 2014 (Soltani, [52]	Randomized, double-blind, parallel	25 participants, per group Age: 48 ± 16	90 mg/per day	4 weeks	Anthocyanin supplementation decreased TC, TG, LDL compared to placebo. >HDL levels.
Kianbakht 2013 [53]	Randomized, double-blind, parallel	40 participants, per group Age: 51.3 ± 15.27	259.68 mg/per day	2 months	Anthocyanin supplementation decreased TC, TG, LDL compared to placebo. >HDL levels.
Schell et al.,2019 [54]	Cross-sectional, randomized, controlled	25 participants (5 men, 20 women) Age: 54 ± 4.2 Obese diabetic adults	225 mg	4 h.	Lower postprandial glucose levels were reported at 2 and 4 h with the anthocyanin group vs. control. IL6/TNF α decreased at postprandial 4 h.
Park et al., 2016 [55]	Cross-sectional, randomized, controlled, double-blind, 4 groups	21 participants (5 men 16 women) Age: 39.8 ± 13.8. Obese adults with insulin resistance	42, 88 and 155 mg	6 h.	The 6-hour postprandial insulin concentration in the group with the highest beverage concentration was significantly reduced.
Richter et al., 2016 [56]	Cross-sectional, randomized, controlled, double-blind,	30 participants (17 men 13 women) Age: 28 ± 2 Healthy, overweight, and obese adults	163 mg	4 h	The intervention did not improve TG, glucose, or insulin.
Abubakar et al., 2019 [57]	Cross-sectional, randomized, controlled, double-blind,	25 male participants. Age: 49 ± 2 Adults: CVD 1–10% in 10 years	150 mg	4 h.	Hibiscus extract improves postprandial vascular function, useful in reducing endothelial dysfunction and cardiovascular risk.
Xiao et al., 2019 [58]	Cross-sectional, three-arm, single blind randomized, controlled trial	21 participants (12 men, 9 women) Age: 34 ± 12 With IR	73, 146 mg	24 h	Significant reduction at 2 h in the area under the curve and a reduction in peak insulin and glucose at 2 h.
Castro-Acosta et al., 2017 [59]	Cross-sectional, randomized, controlled, double-blind,	22 participants (13 men 9 women) Age: 45.4 Healthy	150, 300 and 600 mg	2 h	A reduction in postprandial blood glucose, insulin, and incretin secretion was observed.

## Data Availability

Data described in the manuscript, code book, and analytic code will be made publicly and freely available without restriction upon request to the authors.

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
