# Peer review of "Anthocyanins: Potential Therapeutic Approaches towards Obesity and Diabetes Mellitus Type 2"

_molecules, 2023, doi:10.3390/molecules28031237_

Round 1

Reviewer 1 Report

Through this review, the author wants to address about the controlling of Type 2 diabetes using natural remedies basically the anthocyanin which present in various foods as pigments. The authors discussed role of anthocyanin present in pigmented Maize for controlling of T2DM in model studies reported in various literatures and looks very insightful. But more studies/research may be required to justify the claim since the studies have been performed only in preliminary level.

This is highly relevant to the readers because natural remedy of T2DM will be highly significant than other therapeutics due to zero side effects/other beneficial effect by anthocyanin etc. However, scope of this topic is moderate since the authors limited the discussion up to the anthocyanin present in pigmented maize only.

In principle, I think this review is well written and presentation is novel and merit the interest of Molecules journal. My comments to the submitted review are as follow.

1. Should incorporate heading and subheadings.

2. Since the title of the paper restricts about the discussion of Anthocyanins in Pigmented Maize, the author needs to discuss more about the same but not more about other sources of anthocyanin.

3. In abstract section, incorporation of results of model studies is advisable.

4. The table 1 data represents the various clinical studies about the role of anthocyanin for inhibition of T2DM. Results looks satisfactory but the author should elaborate properly about the all clinical studies in the discussion section.

5. The doses of anthocyanins include a very wide range (32 mg to 600 mg). Is there any rational behind it?

6. The proposed mechanism of inhibition should be discussed in a schematic pathway.

7. Anthocyanin biosynthesis should be presented in a schematic diagram.

8. A description of Figure 3 may be added.

9. The authors should incorporate the reference number against the description in addition with authors name and year (e.g. Chen Y-F, 2016).

10. Although present review emphasis the importance of anthocyanins in pigmented maize, it is suggested to mention importance of modern medicine in relation to efforts towards development of antidiabetic medicine. In this regard, it is recommended to emphasis the importance of iminosugars and sugar derivatives as an antidiabetic agents, and it is suggested to cite following relevant articles related in introduction section.

i.     Nash, R. J.; Kato, A.; Yu, C-. Y.; Fleet, G. W. J. Iminosugars as therapeutic agents: recent advances and promising trends. Future Med. Chem20113, 1513−1521.

ii.     Yang, L.-F.; Shimadate, Y.; Kato, A.; Li, Y.-X.; Jia, Y.-M.; Fleet, G.W.J.; Yu, C.-Y. Synthesis and glycosidase inhibition of N-substituted derivatives of DIM. Org. Biomol. Chem. 2020, 18, 999–1011.

iii.     Chennaiah, A.; Dahiya, A.; Dubbu, S.; Vankar, Y. D. A Stereoselective Synthesis of an Imino Glycal: Application in the Synthesis of (−)-1-Epi -Adenophorine and a Homoiminosugar. Eur. J. Org. Chem2018, 6574−6581.

iv.     Chennaiah, A.; Bhowmick, S.; Vankar, Y. D. Conversion of glycals into vicinal-1,2-diazides and 1,2-(or 2,1)-azidoacetates using hypervalent iodine reagents and Me3SiN3. Application in the synthesis of N-glycopeptides, pseudo-trisaccharides and an iminosugar. RSC Adv20177, 41755−41762.

v.     Rajasekaran, P.; Ande, C.; Vankar, Y. D. Synthesis of (5,6 & 6,6)-oxa-oxa annulated sugars as glycosidase inhibitors from 2-formyl galactal using iodocyclization as a key step. ARKIVOC 2022, vi, 5−23.

Overall, after addressing the points mentioned above, I recommend this review to publish in Molecules.

Author Response

Thank you for your feedback. Please see attached file for punctual adjustments as per your request.

Reviewer 2 Report

Figures 1 and 2 are loose. It would be good to reference them in the relevant paragraph.

 Review the conclusions; some lines seem more like an analysis of results (figure 3 ?). Check out.

Table 1. ? 

Table 1, is loose. It would be good to reference in the relevant paragraph.is loose. It would be good to reference in the relevant paragraph.

(please see the attachment.)

Author Response

(The authors gave the same response as above.)
